# Big Data Analytics and Sensor-Enhanced Activity Management to Improve Effectiveness and Efficiency of Outpatient Medical Rehabilitation

**DOI:** 10.3390/ijerph17030748

**Published:** 2020-01-24

**Authors:** Mike Jones, George Collier, David J. Reinkensmeyer, Frank DeRuyter, John Dzivak, Daniel Zondervan, John Morris

**Affiliations:** 1Virginia C. Crawford Research Institute, Shepherd Center, Atlanta, GA 30309, USA; george.collier@shepherd.org (G.C.); john.morris@shepherd.org (J.M.); 2Center for the Neurobiology of Learning and Memory, University of California, Irvine, CA 92606, USA; dreinken@uci.edu; 3Department of Surgery, Duke University, Durham, NC 27708, USA; frank.deruyter@duke.edu; 4Pt Pal, Altadena, CA 91001, USA; john@ptpal.com; 5Flint Rehabilitation Devices, LLC, Irvine, CA 92614, USA; dzondervan@flintrehab.com

**Keywords:** mobile rehabilitation, disability, rehabilitation, information and communication technology

## Abstract

Numerous societal trends are compelling a transition from inpatient to outpatient venues of care for medical rehabilitation. While there are advantages to outpatient rehabilitation (e.g., lower cost, more relevant to home and community function), there are also challenges including lack of information about how patient progress observed in the outpatient clinic translates into improved functional performance at home. At present, outpatient providers must rely on patient-reported information about functional progress (or lack thereof) at home and in the community. Information and communication technologies (ICT) offer another option—data collected about the patient’s adherence, performance and progress made on home exercises could be used to help guide course corrections between clinic visits, enhancing effectiveness and efficiency of outpatient care. In this article, we describe our efforts to explore use of sensor-enhanced home exercise and big data analytics in medical rehabilitation. The goal of this work is to demonstrate how sensor-enhanced exercise can improve rehabilitation outcomes for patients with significant neurological impairment (e.g., from stroke, traumatic brain injury, and spinal cord injury). We provide an overview of big data analysis and explain how it may be used to optimize outpatient rehabilitation, creating a more efficient model of care. We describe our planned development efforts to build advanced analytic tools to guide home-based rehabilitation and our proposed randomized trial to evaluate effectiveness and implementation of this approach.

## 1. Introduction

Over years of training and experience, a seasoned clinician internalizes an “algorithm” for managing patients based on their presenting characteristics (e.g., type and severity of impairment, reported pain or other complications, baseline fitness level, socioeconomic status, presumed competence in self-management, and “intangibles” such as resiliency, drive, and desire for recovery). By and large, there is a clinical pathway that will be followed for a given impairment or therapy goal, with an established progression of exercises/interventions based on progress made by the patient. The clinician uses an internal “algorithm” derived from years of experience to prescribe the course of therapy, estimate the expected rate of progression, and determine factors that may trigger a course correction. Factors like the level of pain tolerance, need to challenge the patient, and rate of progress may call for a faster or slower progression in the course of exercise. But how does the clinician determine the need for these course corrections? In conventional outpatient practice (Figure 1, orange-colored loop), the clinician must rely on information provided during the clinic visit, such as the patient’s self-reported adherence to prescribed home exercise, reported pain or other problems interfering with exercise, and observed progress (i.e., demonstrated functional capacity in the clinic). Adjustments to the exercise prescription must wait until the next face-to-face visit between the patient and clinician.

Information and communication technologies (ICT) offer another option—digital data collected about the patient’s adherence, performance and progress made on home exercises could be combined with additional clinical information about the patient to help guide course corrections between clinic visits (Figure 1, blue-colored loop). Such data are becoming increasingly available with the advent of app-based therapy management and sensor-based exercise systems [1]. For example, the results of the largest trial of telerehabilitation after stroke were recently published, and a system that incorporated sensor-based games for arm exercise, stroke education quizzes, and videoconferences was shown to be as effective as traditional in-clinic rehabilitation [2]. If the data acquired by apps and sensors are actionable, they could be used to augment information gained during clinic visits to improve the effectiveness and efficiency of outpatient rehabilitation. Indeed, in the telerehabilitation trial, therapists adapted game choices or difficulty parameters during on-line interviews but could also do so off-line based on acquired data. We call this approach—employing ICT to support rehabilitation at home/in the community—mobile rehabilitation or mRehab for short.

A significant barrier to adoption of mRehab by clinicians is the amount of non-reimbursed time and effort required to incorporate new tools and strategies into existing clinical practice. This is particularly nettlesome when their use increases the amount of non-face (and un-billable) time/effort required to manage the patients’ progression. To be readily adopted, mRehab should not increase the amount of time required by the clinician between outpatient visits (e.g., reviewing the patient’s remote sensor data or home exercise log, responding to communications from the patient, or adjusting the prescribed level, intensity or type of home exercise). The real promise of a “smart” mRehab intervention platform is the potential it offers for autonomous or semi-autonomous management of the patient’s progression between outpatient clinic visits.

We believe this promise can be realized by “externalizing” the clinical algorithms successful clinicians use to manage their patients through the course of rehabilitation. Advanced analytic methods brought into prominence by big data applications can be applied to identify patterns in the data collected about patients’ exercise performance and functional activity at home. Combined with additional information about the patient and “algorithmic rules” derived from clinicians’ responses to patient performance (i.e., course corrections from the typical clinical pathway), these data can be used to manage the progression of therapy between clinic visits, provide more customized care based on a patient’s “profile”, and detect/correct the trajectory of recovery to achieve better outcomes. This approach builds on the promise of applying the multitude of recent breakthroughs in big data analytics (BDA) and automated reasoning (e.g., machine learning, deep learning) to support patient care.

In this article, we describe an ambitious project being undertaken by a partnership of researchers, clinicians, and industry partners to explore BDA in medical rehabilitation. We propose and describe a specific mRehab strategy that goes beyond one-on-one telerehabilitation. We call this approach sensor-enhanced activity management (SEAM), enabled by BDA. The goal of the project is to demonstrate how BDA-enabled SEAM can improve the effectiveness and efficiency of outpatient rehabilitation for patients with significant neurological impairment (resulting from stroke, traumatic brain injury, and spinal cord injury). We begin with an overview of BDA and an explanation of how it may be used to optimize outpatient rehabilitation and create a more efficient model of care. Next, we describe our planned development efforts to build advanced analytic tools to guide home-based rehabilitation using a SEAM platform. We conclude with a description of a proposed, hybrid effectiveness trial to evaluate effectiveness of BDA in mRehab and identify factors that influence its application.

## 2. Materials and Methods

### 2.1. Big Data and the Emergence of Advanced Analytic Tools 

The era of BDA arose in response to the explosion of data created by the rise of the Internet. Suddenly, almost overnight in historical terms, a huge collection of computer-readable data was created. It is estimated that, in 1986, 2.6 exabytes (1 exabyte = 1 billion gigabytes) of analog data and only 0.02 exabytes of digital data existed in the world. By 2007, 21 years later, analog data had grown to 19 exabytes but digital data had exploded to 280 exabytes—14,000 times as much data [3]. New approaches to capturing, managing and analyzing this magnitude of data were needed and the field of BDA was born, led by pioneering Internet companies such as Google and Amazon.

Big data is a term used to describe data sets that are so large and complex they cannot be analyzed using traditional data processing applications. While the term big data is relatively new, the act of gathering and storing large amounts of information for eventual analysis is not. The concept gained momentum in the early 2000s when industry analyst Doug Laney articulated the now-mainstream definition of big data as the three “Vs”: Volume—data sets so large that storing and querying the data was extremely difficult—for example the contents of all the world’s websites. Velocity—data streams in at an unprecedented speed and must be dealt with in a timely manner. RFID tags, sensors and smart metering are driving the need to deal with torrents of data in near-real time. Variety—data comes in all types of formats—from structured, numeric data in traditional databases to unstructured text documents, email, video, audio, stock ticker data and financial transactions [4]. The emergence of big data as a field, was driven by collecting data from a variety of sources, including business transactions, social media and information from sensor or machine-to-machine data. In the past, storing and processing such large, heterogenous or “fast” data sets was a daunting task—but a multitude of new technologies (such as scale-out massively parallel systems e.g., Hadoop and Apache Spark; NoSQL databases, elastic cloud computing and many others) have eased the burden.

Big data analysis often requires specialized analytic tools (e.g., data visualization, artificial intelligence, machine learning) that can detect patterns, trends and correlations in the data not evident using traditional approaches. These patterns and trends can be used to identify important relationships that may, in turn, be used to make predictions. In this regard, big data may be thought of as any voluminous amount of information that can be mined for useful, actionable information [5]. Our project will leverage some of the key resulting technologies that the late 20th century data explosion demanded.

To that end, we see many advantages of using a big data architecture such as the one we describe in some detail below. This architecture is based on a “data lake” with an overlay of massive parallel processing analytics and will run in the cloud. A data lake is an extremely scalable, storage system which provides redundant storage to achieve availability and reliability. Data in the “lake” is typically stored in a raw format and organized into a “flat” structure with a set of tagged, partitioned files. A scalable computational layer above the lake supplies parallel computing engines which can query and process the data stored in the lake using both traditional and nontraditional techniques. Systems such as these were created to address a variety of issues that arose during the data explosion.

An important technology set that has reemerged to create value from these large data sets is machine learning (ML) and artificial intelligence (AI). These technologies already have a large impact on our lives. ML and AI are in extensive use in automobiles, genetics, medicine, finance, etc., to automate procedures, reduce processing time, and remove the possibility of human errors. ML helps in analyzing at scale, thus helping make quicker and better decisions. ML is also making important inroads into medicine. For example, JAMA has a website specifically dedicated to advances in applying ML [6]. According to Google Scholar there have been more than 185,000 scientific publications on ML applications in medicine in the last five years (less than 20% reference medical rehabilitation).

A Google AI algorithm identified diabetic blindness using fundus photographs at a high level of accuracy. The algorithm identified retinopathy at a sensitivity of 96.1% and specificity of 93.9% [7]. ML has been used to predict adverse drug reactions [8]. Clinical pathways for chronic disease care have been created using ML algorithms. This system can divide patients into risk-based sub-groups [9]. The progression of chronic hepatitis C virus was predicted using boosted tree survival models [10]. A deep learning algorithm was 100% effective at identifying invasive forms of breast cancer from pathology images [11]. ML led to substantially improved discrimination and calibration for predicting the risk of emergency admissions [12]. A ML model could use biometric data monitored in the intensive care unit (ICU) to suggest types of treatments needed for different symptoms [13].

Past work has applied ML models such as decision trees, hidden Markov models, logistic regression, support vector machines, and random survival trees to efforts to improve patient adherence to treatment. For example, Lee and colleagues [14] used a support vector machine to predict medication adherence in elderly patients with chronic diseases with an Area Under the receiver operating Curve (AUC) of 97%. Self-reported self-efficacy was the most significant contributor to prediction. A study of Parkinson’s patients [15] leveraged data from sensors that measured whole-body movement. The authors were able to detect medication adherence by applying a support vector machine model. They achieved an accuracy of 95% for patient-specific models and an accuracy of 78% for a generalized model for the population. In a more complex multistage analysis, the application of random survival forests allowed the derivation of empirical medication adherence thresholds predictive of hospitalization risk [16]. A review of the literature on ML and cancer treatment found that ML techniques such as neural networks, Bayesian networks, support vector machines, and decision trees have proven so successful in cancer treatment they are now widely applied in academic treatment settings [17].

We believe this is an opportune time to apply BDA and predictive ML models in mRehab. And others have also emphasized the rise of big data opportunities and challenges in rehabilitation medicine [18,19]. Until recently, rehabilitation data were comprised primarily of medical records updated at visits to medical providers. However, the advent of mobile-device-based therapy management programs and sensor-guided home exercise is ushering in a new era of data characterized by volume, velocity, and variety. This leads to the possibility of generating predictive ML models to adjust home-based therapy programs as proposed in Figure 1. For example, clustering (e.g., hierarchical clustering, k-means clustering, etc.) and classification models (e.g., logistic regression, random forest) can be applied to mathematically group patients into different categories to select their optimal rehabilitation pathway and expected rate of progression. Machine learning clustering algorithms have already been shown to be a better predictor of rehabilitation potential than a standard clinical assessment protocol [20] and AI can successfully learn clinical assessment of spasticity with the accuracy of clinical assessors [21].

Similar strategies have been used in management of patients with diabetes and brain tumors [22,23]. Automated clustering will help show us if there is an underlying structure that matches the clinician’s internal algorithm, such that the machine can learn how to group patients into predictive categories. Once determined these categories could be used as predictors themselves or incorporated into the clinician’s decision making.

As for the trajectory of rehabilitation progress, we can apply techniques from the ML subfield of process mining [24]. Process mining techniques can be used to perform “next task prediction” using random forest algorithms [25] or deep learning algorithms [26]. Process mining can also be used to predict the processes’ final state. For instance, it has been applied to predicting cancer progression and outcomes [17]. Other fruitful techniques are likely to come from the disciplines of time series analysis [27] and survival analysis [28].

### 2.2. Development Efforts 

The project includes three development efforts: (1) create a large “data lake” repository for data generated on medical rehabilitation outpatients by our four partner organizations in this effort (Shepherd Center, UC-Irvine Medical Center, Pt Pal and Flint Rehab), (2) merge the functionality of two commercially-successful therapy management platforms (Pt Pal and Flint Rehab) to create a sensor-enhanced activity management (SEAM) platform for prescribing and tracking use of therapeutic activities/exercises at home, and (3) produce and deploy useful analytic techniques for patient management, and, in particular, matching patients to strategies that maximize adherence, engagement, and outcomes of home-based therapeutic interventions.

Our proposed big data architecture leverages powerful, rapidly evolving, yet robust and secure, Internet driven technologies. We will take advantage of cloud computing, business intelligence systems, massive data storage, data mining, ML and AI tools. The proposed management tools will leverage ML and AI to alert clinicians to patient behavior that is trending in the wrong direction. Additionally, we will apply time series analysis to develop predictive models of therapeutic outcomes that can take advantage of the person-specific home and clinical data available, as well as a large collection of historical data to improve outcome predictions. Once trained, such models can be used, with input of ongoing remote sensing data on home exercises, to plan and course-correct therapy.

### 2.3. Technological Foundation 

Recent advances in home therapy technologies provide a means to gather the patient performance data needed to make real-time therapy decisions. In this project, we will use (1) app-based therapy management (Pt Pal) and (2) sensor-based, activity tracking and gamified exercise (FitMi) (Figure 2).

Pt Pal (Figure 2, top row) is a cloud-based patient engagement platform that enables clinicians to manage patients between visits at home by sending treatment plans, therapies, activities and reminders to the patient’s mobile device. Pt Pal also supports clinical assessments, compares results against other outcomes data, provides active reminders, and incorporates both stock and custom video examples. FitMi (Figure 2, bottom row) is a sensor-based solution for gamification of home-based rehabilitation exercise. FitMi consists of two hockey-puck-like devices (“motion interfaces”) that sense force and motion and connect wirelessly to a PC or mobile device. The FitMi software (called “RehabStudio”) guides users through a library of 40 exercises designed by experienced rehabilitation therapists, including exercises for the hand, arm, core, and legs, providing motivating feedback and adaptively increasing challenge based on performance. Both technologies are growing in their usership: Pt Pal has been used by over 500,000 patients since its introduction in 2013, and the number of FitMi users has grown to over 5000 since Flint introduced FitMi in 2017, with over 70 million practice movements logged in RehabStudio. In terms of feasibility for widespread dissemination, Pt Pal costs $15 to $100 per year per clinician (for use with an unlimited number of patients) depending on the number of clinicians in a healthcare setting using the system. FitMi is currently sold for $299.

Pt Pal is a robust platform which enables clinicians to send prescribed exercise plans to their patients/caregivers’ mobile device and collect adherence data. The patient/caregiver works from their daily task list of exercises and activities, documenting either through a count-down method or marking functional activities (e.g., getting dressed) as completed. Pt Pal collects subjective data regarding the quality or quantity of exercises performed and does not gamify exercises. The FitMi is a stand-alone product which motivates users to perform self-guided exercises with engaging games that automatically adapt to the user’s ability but does not provide a closed-loop solution for connecting patients—and their exercise data—back to their care providers. Our project will combine these technologies into a single system to get the benefits of both. The resulting system will allow development of a first-of-its-kind therapy management platform and BDA toolkit for better understanding and optimization of outpatient medical rehabilitation outcomes.

Our initial focus is on outpatient rehabilitation of individuals with physical impairments due to stroke and spinal cord injury because these are the largest potential user populations, and the existing, sensor-enhanced mRehab tools supported on the Flint Rehab platform are for physical rehabilitation. However, we foresee that BDA-enabled SEAM platforms will also be useful for people with other neurological conditions, including traumatic brain injury, cerebral palsy, and multiple sclerosis, as well as orthopedic injuries and other neuromuscular diseases. We will develop our SEAM system to be a general-purpose rehabilitation platform, customizable to different therapy needs by adding activities and exercises to its library. To this end, additional development efforts include mRehab tools to address sensorimotor, emotional, cognitive and other impairments that will be relevant to many patient populations.

### 2.4. Data Collection and Analytics 

Central to the project we are proposing is the collection of a variety of data from partnering organizations. We will fill the data lake with historic, de-identified data from past and existing patients. These data currently represent over 500,000 patients and millions of therapy sessions, and will provide a rich source of information for formative ML and AI applications.

We have organized the data lake and analytics infrastructure into layers (Figure 3). The human and system actors interacting with this infrastructure are shown in Figure 4. We based this layering on a typical enterprise data architecture with some adjustments to efficiently serve the project’s primary purposes. Our project will deliver managed data sharing, data exploration, modeling, and advanced analytics application development.

Our first layer is for data ingestion. We provide at least two mechanisms for data transfer and ingestion. For simple scenarios, we provide a dedicated SFTP (secure-file-transfer-protocol) virtual machine with an administration interface. For more complex scenarios, we provide Azure Data Factory, a robust cloud-based, serverless ETL (extract/transform/load) tool with a graphical programming interface. Apache Spark is integrated into this tool for challenging ETL jobs if needed.

The data ingestion layer is integrated with an inexpensive Azure scale-out storage solution called Azure Data Lake Storage Gen2. This data storage layer is optimized for Analytics providing massive scaling, data lifecycle policy management, and high availability. The Data Lake layer delivers Hadoop compliant file storage with atomic file and folder operations. For security purposes, a fine-grained access control system is integrated into overall security. We need not worry about data size as storage scales to a massive size if needed, and there are no limits on file sizes or numbers of files.

Importantly, the data layer is integrated into the next layer, which is for querying and processing. We provide two other robust systems to process data and provide querying capabilities. The first system is Apache Spark, which grew out of the analytics communities’ experience with Hadoop. Like Hadoop, Spark is a cluster computing system which scales computing by adding more compute nodes and cores across multiple machine instances. Unlike Hadoop, Apache Spark executes in distributed memory, making it exceedingly faster than Hadoop for suitable jobs. While Spark does not have the large ecology of applications and programming that Hadoop does, it excels at interactive querying and machine learning workloads, which are two prime use cases for the project. Spark provides better ease of use than Hadoop and has APIs (application programming interfaces) for Scala, Java, and Python. It also provides an SQL-like API called Spark SQL.

For extremely complex data processing and querying, we can leverage a different and very rich system that has a SQL and relational flavor. This system is Azure Synapse Analytics, which can operate across SQL databases and files in the Data Lake. It integrates Apache Spark and the SQL engine and can integrate queries across both. Multiple programming languages are supported, including T-SQL, Python, Scala, Spark SQL, and .Net.

Above the data processing and querying layer is the analytics product development layer. A primary goal of the analytics effort is to develop predictive machine learning models. To that end, a variety of machine learning, visualization, and data exploration tools are integrated into the querying and processing layer. These tools include Apache Spark and Azure Machine Learning. Apache Spark has built-in machine learning capabilities. While not comprehensive, these capabilities can scale to vast data sets. Azure Machine Learning (ML) is a diverse environment with many capabilities that provides an end-to-end solution to develop and deploy ML models. Azure ML takes a “dev-ops” approach, bringing together a suite of capabilities from version management, machine experiment management, ML pipelines, model validation, and deployment. The system wraps ML development by integrating access to compute resources and managing and securing the models. Many ML projects are very successfully carried out on individual developers’ workstations; however, the ML tools mentioned create shared development environments. This is an important consideration for the project.

Data will be collected moving forward on all patients who are prescribed home-based therapies between outpatient visits. In addition, we will include relevant data from the electronic medical record (EMR) for Shepherd Center and UC Irvine patients who receive outpatient therapy. We will harvest data for storage and analytics as it is being generated, combining new data with historical data.

Our goal is to collect as much data as is appropriate and possible. The data will be de-identified for study and stored in a portion of the Azure HIPAA-compliant cloud environment. Azure’s Blob Storage will be used as our data lake repository. This is an inexpensive storage solution compatible with a variety of data formats and capable of expanding to very large data sets. Blob storage is priced according to usage and we expect that this component makes only a small contribution to our IT costs.

Preliminary queries of the data lake will attempt to learn more about the progression of outpatient therapy for different patients and presenting problems. We will attempt to identify factors associated with successful adherence to home-based exercises and progression in recovery. Initial questions of the data will include: (1) how many home exercises do therapists typically prescribe and are there standard sequences of exercise through which they progress patients; (2) what factors do therapists evaluate to determine when and how to update exercise parameters (e.g., number of repetitions or intensity of exercise); (3) how often do therapists update the parameters of prescribed exercises; (4) what is the relationship between patient adherence and the number of exercises prescribed; (5) how often do therapists check on patient progress between clinic visits; and (6) what patient and therapy factors are predictive of patient adherence and progress? Information gleaned from these and additional follow-up questions will be used to guide development of therapy management tools that can be used to progress patients between outpatient clinic visits.

Although the final mRehab management tools will be determined by our development effort, a simplistic example of how they might work follows. Patients may fit into one of four behavioral profiles: (1) nonadherent in completing home exercises as prescribed, (2) adherent with the exercise program but not making progress on desired therapy goal, (3) on track in completing exercises as prescribed and making progress on therapy goal, and (4) over-performing on exercise program and progressing too fast for optimal recovery (also at risk for injury and dissatisfaction with pace of progressing exercises). Our big data analysis will likely identify more behavioral profiles—or sub-profiles—and the factors that predict which profile a patient’s behavior is likely to follow. For example, different patients will respond differently to different progression strategies, including different types and frequencies of feedback.

The therapy management algorithm will include the typical “clinical path” of exercise progression and frequency of clinic visits, and has three basic elements that can be adjusted in response to the patient’s profile: (1) the pace of progressing exercises (e.g., varying intensity or type of exercise), (2) messaging from the “therapist” (can be pre-programmed for delivery, or system could also prompt therapist to call or email the patient with a custom message) that includes “nudges” to prompt adherence or performance, social reinforcement for desired performance, and warnings concerning adverse outcomes (injury from overdoing exercise), and (3) the frequency or timing between outpatient visits (could be increased or decreased based on progress or problems observed). Clinicians may choose to use information generated by these management tools or use a semi-autonomous mode, wherein the clinician sets the parameters for modifying patient progression, messaging, and scheduling of outpatient visits, and the software executes patient management accordingly.

### 2.5. Evaluation of Project Effectiveness 

We will conduct a Type 1 hybrid, randomized controlled trial (RCT) [29] to evaluate effectiveness and implementation of an alternative model of care utilizing mRehab interventions to optimize outpatient therapy. This trial will involve cluster randomization by clinician, so that each participating clinician will have patients randomized to one of three groups: (1) patients who receive conventional outpatient therapy (“usual care”), (2) patients who are managed by the clinician using information provided by the mRehab management tools (patient profiles and algorithm-supported therapy management between visits) developed as part of our development project, and (3) patients whose progression is managed in semi-autonomous mode by the therapy management tools developed.

For patients randomized to group 2, clinicians will have access to patient profiles that suggest an optimal progression of therapy and management scenario (e.g., frequency of outpatient visits, frequency and type of messaging between visits and in response to adherence and engagement with home-based therapy). They will also have the option to use programmed clinical algorithms to adjust a patient’s therapy based on his or her actual adherence, engagement, and progress as measured by instrumented mRehab tools. For patients randomized to Group Three, the clinician will be asked to set the parameters for management and system algorithms will execute the plan accordingly, notifying the clinician of actions taken.

We will evaluate differences between groups in the rate of progress on targeted clinical outcomes. Our hypothesis is that participants randomized to Groups Two and Three will demonstrate a faster trajectory of improvement because of more efficient management of their home exercise programs between visits. Non-inferiority of Group Three to Group Two would verify effectiveness of semi-autonomous management of the patient’s progression in therapy. We will also collect secondary outcomes data on satisfaction of patients and providers with use of the mRehab management tools (in clinician-controlled or semi-autonomous mode), and on amount of time spent by clinicians on patient management between outpatient clinic visits.

We will examine between-group differences in: (1) changes made in therapy progression, communications, and latency between clinic visits, (2) patient adherence, performance, functional progress, and achievement of therapy goals, (3) number of clinic visits and time required to achieve therapy goals (i.e., faster or further progress using mRehab management tools), and patient and clinician satisfaction with therapy delivery and outcomes, and time/effort required for patient management. We will also gather input from clinicians involved in the study and clinic management about ease or difficulty (facilitators and obstacles) of incorporating mRehab strategies into the clinical workflow and current clinical management practice.

For the proposed hybrid RCT, we will use a repeated-measures analysis technique such as growth curve analysis (GCA) [30] to determine significant changes over time in targeted therapy outcomes and effects of group assignment (usual care, use of mRehab management tools by clinician, and semi-autonomous management). Primary outcomes data will be collected at the first outpatient visit, periodically throughout therapy, and at the last visit. Although the frequency of visits and duration between visits will vary by patient, we will be able to track multiple data points (repeated measures) for each patient and will examine data over the first six months of therapy. GCA estimates initial status and growth over time and variance around the two main parameters—targeted outcome and group assignment. Initial status coefficients (regression constant) allow us to see patient outcome status at the beginning of therapy (first point in time), and growth rate coefficients show how function changes (grows) over time on the targeted outcome. GCA estimates both fixed and variable effects in multivariate datasets. Fixed effects refer to the average regression coefficients for all subjects in each group. Variable effects refer to the variance around regression coefficients for each observation—how much each group varies from the regression coefficient. GCA is advantageous for datasets like ours, as it accounts for missing data when estimating growth parameters [31].

An exploratory heterogeneity of treatment effects (HTE) analysis [32] will examine differences in intervention effectiveness for different patient types (SCI, stroke), behavioral profiles (e.g., non-adherent, fully adherent), and therapy goals (e.g., upper extremity or lower extremity strength, range of motion, etc.). First, we will examine mean differences across all outcome measures between each of the three treatment groups to identify differences in outcomes. Second, we will add an interaction term to our GCA to determine interaction effects between therapy, patient type, behavioral profile, and management approach (group assignment). One assumption we can explore with this HTE analysis is whether the different management approaches have different effects for different patient behavioral profiles.

Results from this trial will determine if use of mRehab strategies for autonomous management of outpatient therapy is more effective and efficient than conventional outpatient therapy. Feedback from patients and therapists about acceptability of and satisfaction with this approach will support adoption and encourage sustainability of this new model of care.

## 3. Conclusions

Societal trends are converging that will increase the demand for and expected cost of rehabilitation services. These demands are likely to accelerate the shift from inpatient to outpatient venues of care [33]. In addition to lower overall cost, outpatient care offers the potential advantage of tailoring therapies to the home and community setting where the patient resides. However, the advantages of outpatient care are offset by several challenges to effective service delivery. Frequent problems include measuring gains of daily functioning over time, assessing the effects of timing and dose of interventions related to patient functioning, assessing adherence with instructions for home-based therapeutic activities, providing effective feedback to patients about performance of therapeutic activities, and being able to update instructions more frequently than only at the time of an outpatient visit.

In this article, we have attempted to make the case for the application of BDA-enabled SEAM platforms to improve efficiency and effectiveness of conventional outpatient rehabilitation. Technologies that have already transformed communication, entertainment and business are poised to create the same transformation of rehabilitation services for people with neurological impairment. We provide a roadmap for one probable course of this transformation, although there are still many uncertainties in this roadmap. For example, we don’t know yet which data derived from which sensors and apps will be most actionable, how best to reduce that data to present the most actionable information to therapists and patients, what degree of automation therapists will accept for what types of patients, and the degree to which implementing different therapist algorithms will affect patient performance. Yet, by making progress along this roadmap, we will implement the infrastructure and acquire the data needed to rigorously answer these questions.

## Figures and Tables

**Figure 1 ijerph-17-00748-f001:**
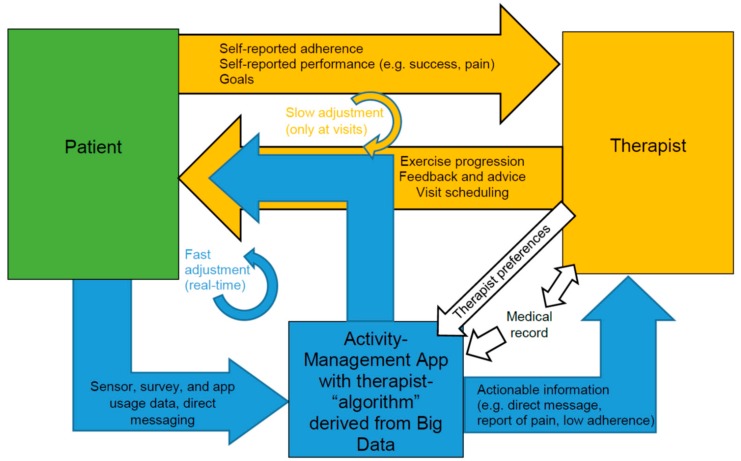
Mobile rehabilitation (mRehab) using sensor-enhanced activity management (SEAM). The orange-colored loop represents the conventional approach to adjusting home exercise programs. The therapist interviews the patient and performs assessments at clinic visits, then adjusts exercise progression and provides feedback and advice. A SEAM platform implements the blue-colored loop, using sensor data to more objectively track patient adherence and performance in real-time. Large amounts of unreduced data are of little value to therapists, so we propose that it is essential for SEAM platforms to use big data analytics (BDA) to develop a therapist-like “algorithm” that can adjust exercise progression and provide simplified feedback to the patient and therapist (via an activity-management app), consistent with the therapist’s and patient’s preferences and intentions. Our working hypothesis is that adding the BDA-enabled SEAM loop to the conventional adjustment loop will increase patient compliance and enable more optimal delivery of home rehabilitation.

**Figure 2 ijerph-17-00748-f002:**
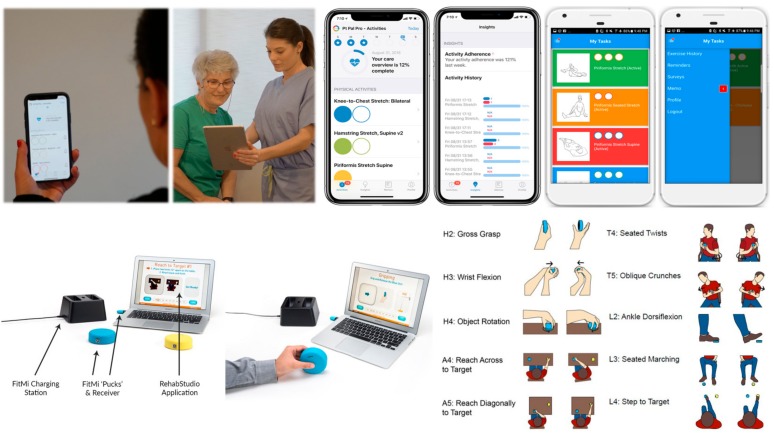
Top row: Pt Pal (produced by the company Pt Pal) consists of an app for patients and a portal for clinicians. The app replaces paper given by a therapist or doctor, and can convey activities, exercise instructions, surveys, and education (see screenshots). The portal allows clinicians to configure individualized protocols, automate communication, and monitor outcomes. Bottom row: FitMi (produced by the company Flint Rehab Devices) consists of two force and motion sensing pucks and a software application called RehabStudio. Users can choose from 40 exercises of varying difficulty, for the hand, arm, core, or legs. New, more challenging exercises are automatically unlocked as people achieve goal intensities for easier exercises. This project will combine the activity management of Pt Pal with the sensing and gamification of FitMi to promote engagement and monitor progress in home rehabilitation.

**Figure 3 ijerph-17-00748-f003:**
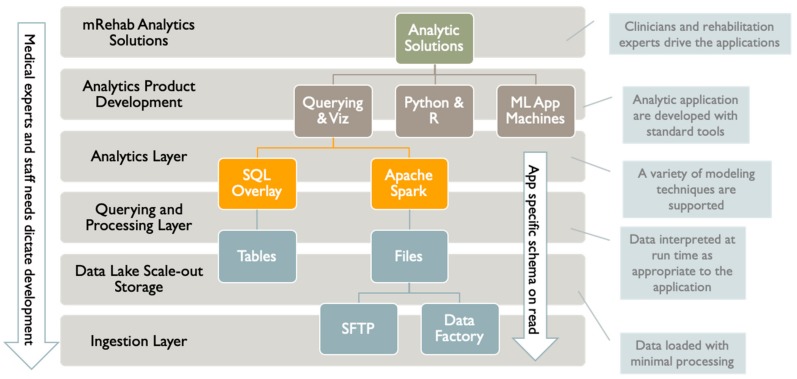
Layered Analytics Infrastructure.

**Figure 4 ijerph-17-00748-f004:**
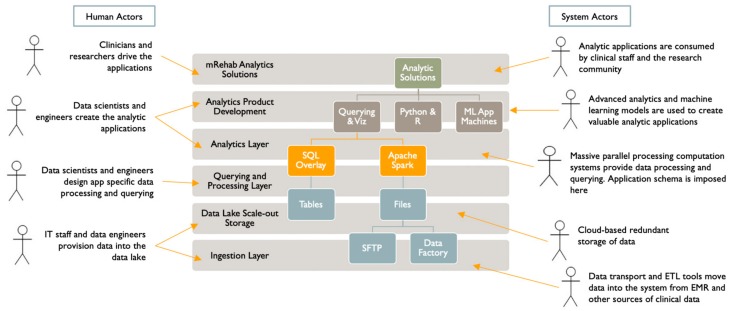
Human and System Actors.

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
