# Peer review of "Big Data Analytics and Sensor-Enhanced Activity Management to Improve Effectiveness and Efficiency of Outpatient Medical Rehabilitation"

_ijerph, 2020, doi:10.3390/ijerph17030748_

Round 1

Reviewer 1 Report

The authors describe a conceptual approach towards the development of a Sensor-Enhanced Activity Management (SEAM) platform for prescribing and tracking the use of therapeutic activities/exercises at home and in the community development. The platform will include a data lake repository for data on medical rehabilitation outpatients and will provide data analytics functionalities for the analysis of the huge amount of data to maximize the adherence, engagement, and outcomes of home- and community-based therapeutic interventions. Minor English editing is required. The authors mention that the platform will be based on the integation of two commercially-succesfull platforms and thus its innovation is obscured. The big data analytics functionalities are not clearly presented and more information should be added. A more technical figure should be added to clearly demonstrate the conceptual architecture of the platform in terms of actors, processes/workflows and output. Use cases should also be provided along with a section regarding the sustainability of the platform in prospective studies.

Author Response

Thank you for your review and useful feedback on our manuscript. In response to your recommendations, we have 1) provided a more detailed description of the big data analytics functionalities we propose, including an additional figure (3) showing the multi-layered analytics infrastructure (pp 5-8); 2) provided more details about how the mRehab therapy management tools might be deployed and evaluated in practice compared to conventional outpatient therapy (pp 8-10); and commented on how results of our hybrid effectiveness-implementation trial, if successful, will support adoption and sustainability of this new approach lines 447-450).

Reviewer 2 Report

The authors has proposed a sensor-enhanced home exercise and big data analysis framework for medical rehabilitation. The proposed method may have some merits, but some major concerns are:

1. The major problem of the work is that the quantitative comparison is missing. How can we say the proposed framework is more effective?

2. The literature review of the study is insufficient, there are many important studies that are missing. The authors should include the following references, e.g., IoT, medical big data analysis, with some proper discussions:

C. Wang et al. "SaliencyGAN: Deep Learning Semi-supervised Salient Object Detection in the Fog of IoT," in IEEE Transactions on Industrial Informatics. doi: 10.1109/TII.2019.2945362

Mohammadreza Soltaninejad et al. Automated Brain Tumour Detection and Segmentation using Superpixel-based Extremely Randomized Trees in FLAIR MRI. The International Journal for Computer Assisted Radiology and Surgery. 12(2), 183--203.

3. The other concern is that how the proposed framework can cope with the further clinical studies?

Author Response

Thank you for your thorough review of our manuscript and the recommendation offered for improvement. With respect to your first point, we agree that this concept paper does not include a quantitative comparison.  It does, however, proposed that very comparison which would evaluate effectiveness of the mRehab platform we are developing compared to conventional outpatient therapy, and verify the feasibility of implementation in applied clinical settings.  With respect to the limited literature review, we have added additional discussion about big data analysis in medicine and rehabilitation specifically (lines 157-213). Finally, with respect to the third point about how the proposed framework can cope with further clinical studies, we did not understand this comment or what revisions to the manuscript might be in order to address the comment.

Round 2

Reviewer 1 Report

The authors have addressed most of my comments. The actors of the platform should be added in Fig. 3 (e.g., the actor who provides the data, the actor who processes the data) along with the links with the corresponding layers. In addition, the layers in Fig. 3 should be somehow connected to better understand whether the provided schema is presented in a bottom-up or top-down architecture. Each box in the layer should be clearly explained in the description. The innovation of the platform should be highlighted in Fig. 3 (e.g., what are the analytic soultions and whats the novelty?)

Author Response

Thank you for your insightful feedback concerning our manuscript and suggested changes to Figure 3 (and related text).  We have revised Figure 3, added Figure 4 in order to show the human and system actors as you suggest, and revised the narrative to provide further description of the interactions between the data lake infrastructure, human and system actors.

Round 3

Reviewer 1 Report

The authors have addressed my comments. In its current form, this conceptual paper can be considered for publication.